# Ameliorative Effect of *Citrus junos* Tanaka Waste (By-Product) Water Extract on Particulate Matter 10-Induced Lung Damage

**DOI:** 10.3390/nu14112270

**Published:** 2022-05-28

**Authors:** Wen-Yan Huang, Wan Heo, Inhye Jeong, Mi-Jeong Kim, Bok-Kyung Han, Eui-Cheol Shin, Young-Jun Kim

**Affiliations:** 1Department of Food and Biotechnology, Korea University, Sejong 30019, Korea; flyhighwy@korea.ac.kr (W.-Y.H.); isaac36@korea.ac.kr (I.J.); puppy99@korea.ac.kr (M.-J.K.); hanmoo@korea.ac.kr (B.-K.H.); 2BK21 FOUR Research Education Team for Omics-Based Bio-Health in Food Industry, Korea University, Sejong 30019, Korea; 3Department of Food Science and Engineering, Seowon University, Cheongju 28647, Korea; 01062033526@seowon.ac.kr; 4Department of Food Science, Gyeongsang National University, Jinju 52828, Korea; eshin@gnu.ac.kr

**Keywords:** *Citrus junos* Tanaka, waste, by-product, particulate matter 10, pulmonary, inflammation

## Abstract

*Citrus junos* Tanaka (CJ)-related products are well-accepted by consumers worldwide; thus, they generate huge amounts of waste (peel, pulp, and seed) through CJ processing. Although some CJ by-products (CJBs) are recycled, their use is limited owing to the limited understanding of their nutritional and economic value. The exposure to particulate matter (PM) increases the risk of respiratory diseases. In this study, we investigated the ameliorative effects of CJB extracts (100, 200 mg/kg/day, 7 days) on PM10-induced (10 mg/kg, intranasal, 6 h) lung damage in BALB/c mice. Cell type-specific signaling pathways are examined using the A549 (PM10, 200 μg/mL, 6 h) and RAW264.7 (LPS, 100 ng/mL, 6 h) cell lines. The CJB extracts significantly attenuated PM10-induced pulmonary damage and inflammatory cell infiltration in a mouse model. The essential protein markers in inflammatory signaling pathways, such as AKT, ERK, JNK, and NF-κB for PM10-induced phosphorylation, were dramatically reduced by CJB extract treatment in both the mouse and cell models. Furthermore, the CJB extracts reduced the production of reactive oxygen species and nitric oxide in a dose-dependent manner in the cells. Comprehensively, the CJB extracts were effective in reducing PM10-induced lung injuries by suppressing pulmonary inflammation, potentially due to their anti-inflammatory and antioxidant properties.

## 1. Introduction

*Citrus junos* Tanaka (CJ), also known as yuzu, is a yellow citrus fruit widely grown in Korea, Japan, and China [1]. CJ is well-accepted by consumers because of its pleasant flavors and abundant phytochemicals. CJ has been suggested as a good source of dietary antioxidants, and its role in the prevention and treatment of various diseases has been widely studied [2,3,4]. The major phenolic compounds found in CJ are flavonoids (including naringin, hesperidin, narirutin, and neohesperidin), which are considered one of the most important sources of bioactive compounds [5,6,7]. CJ has been widely used as a beverage, ice cream, dessert, and cooking material because of its tart flavor. However, except for the edible flesh used in the processing industry, the other inedible parts, such as the peels, seeds, and pulp matrices, are mostly wasted as by-products. Although some CJ by-products (CJBs) from the food processing industry are recycled, a large quantity is still released into the environment, causing pollution issues [8,9,10]. Additionally, the use of by-products is limited owing to the lack of understanding of their nutritional and economic value.

Air pollution is a major concern worldwide owing to its impact on human health. Although various pollutants contribute to air quality, particulate matter (PM) is one of the most important determinants of induced biological effects [11]. PM10 (PM with an aerodynamic diameter of under 10 μm) can easily be inhaled and deposited throughout the airways, which harms the human respiratory and circulatory systems [12,13]. As it is mainly absorbed through respiration, many studies related to pulmonary inflammation have been conducted, while reactive oxygen species (ROS)-mediated redox responsive signaling pathways, including phosphoinositide 3-kinase (P13K)/AKT, mitogen-activated protein kinase (MAPK), and nuclear κappa B (NF-κB), have been identified as major mechanisms of action explaining its toxicity [14,15,16]. In addition, PM10-induced pulmonary oxidative stress is associated with the release of various cytokines, such as inducible nitric oxide synthase (iNOS), cyclooxygenase-2 (COX-2), tumor necrosis factor alpha (TNF-α), interleukin-6 (IL-6), and interleukin-1 beta (IL-1β), which can promote lung damage [17,18,19].

The increasing annual prevalence of PM10-induced diseases imposes enormous multidimensional burdens on patients and healthcare systems, necessitating the exploration of newer, safer substances for treatment, such as natural products. Our previous studies [20] demonstrated the potent antioxidant effect of CJ peel extract in PM10-exposed mouse lungs. The purpose of this study was to reuse CJ by-products (CJBs), which have more abundant nutritional bioactives than CJ peel alone, to minimize environmental stress. Thus, we explored whether CJB could alleviate PM10-induced pulmonary inflammation using in vitro and in vivo models.

## 2. Materials and Methods

### 2.1. Materials and Reagents

The CJBs used in this study were obtained from the Goheung Agricultural Cooperative Federation (Goheung, Korea) and extracted through the methods used by Lee et al. [20], with some modifications. The material was ground using a blender and extracted (70 °C) with 10 volumes of distilled water for two hours. The extract was filtered through a filter paper (Whatman, Maidstone, UK), spray-dried, and stored in a −80 °C deep-freezer. The naringin (≥95%), hesperidin (≥97%), and PM10 (ERM-CZ120) used for the certified reference material were purchased from Sigma-Aldrich (St. Louis, MO, USA).

### 2.2. Antioxidant Capacity and Component Assay

#### 2.2.1. ABTS Radical Scavenging Assay

The ABTS radical scavenging activity was estimated using the method of Shalaby and Shanab [21], with some modifications. Briefly, an ABTS solution (190 μL; 1 mM AAPH and 2.5 mM ABTS in PBS; Sigma-Aldrich) was mixed with 10 μL of a CJB (1 mg/mL; distilled water (DW)) solution and allowed to react for 10 min at room temperature in the dark. The absorbance at 734 nm was measured using a Perkin Elmer Lambda 35 UV/VIS spectrometer (Perkin Elmer Inc., Waltham, MA, USA). The radical scavenging activity of the CJB extract was expressed as the ascorbic acid equivalent value (VCE).

#### 2.2.2. DPPH Radical Scavenging Assay

The DPPH radical scavenging activity was evaluated through the method used by Shalaby and Shanab [21], with slight modifications. Briefly, a DPPH (Sigma-Aldrich) solution (190 μL; 50 μM in 95% EtOH; Sigma-Aldrich) was added to 10 μL of CJB (1 mg/mL; 95% EtOH) solution and incubated at room temperature for 30 min in the dark. The mixture was stirred vigorously and was allowed to stand at room temperature in the dark for 30 min. The rate of absorbance at 517 nm was monitored using a spectrophotometer (Perkin Elmer), and the results are presented as the mg VCE/g sample.

#### 2.2.3. Total Polyphenol Content

The total polyphenol content (TPC) in the samples was determined using the method described by Singh et al. [22]. Briefly, 50 μL of the CJB solution (1 mg/mL; DW) was reacted with 100 μL of Na_2_CO_3_ (2% in DW; Sigma-Aldrich) solution and 50 μL of the Folin–Ciocalteu solution (0.2 N; Sigma-Aldrich), blended, and preserved for 10 min. The absorbance was measured at 720 nm using a spectrophotometer (Perkin Elmer), and the results are presented as the gallic acid equivalence value (GAE).

#### 2.2.4. Total Flavonoid Content

The total flavonoid content (TFC) in the samples was determined using the method of Singh et al., with slight modifications [22]. Briefly, the 500 μL CJB solution (1 mg/mL) was mixed with 200 μL NaNO_2_ (10% in DW; Sigma-Aldrich), and the mixture was incubated for 6 min at room temperature. After that, 300 μL of AlCl₃ (10% in DW; Sigma-Aldrich) was added to the mixture and incubated for 6 min again, and then 2 mL of 1 M NaOH was added, blended, and preserved for 15 min at room temperature. The absorbance was measured at 510 nm using a spectrophotometer (Perkin Elmer), and the results are presented as the rutin equivalence value (RE).

### 2.3. Analysis of Naringin and Hesperidin Content

The naringin and hesperidin contents in the samples were determined using the method of Lee et al., with some modifications [20]. The Shiseido Nanospace SI-2 (Shiseido; Tokyo, Japan) was equipped with a quaternary pump, an online degasser, an auto plate-sampler, and a UV/Vis detector with a CapcellPAK C18 UG120 column (4.6 × 250 mm, 5 µm particle size), which was used for the separation. Solvent A consisted of 0.1% formic acid in water, and solvent B consisted of 0.1% formic acid in acetonitrile. The solvent gradient was listed as follows: 15% B at 0–15 min; 15–35% B at 15–35 min; 35–90% B at 35–40 min. The composition was held at 95% B for 5 min and then returned to its initial conditions, and it was maintained for 5 min to equilibrate the column. The detection wavelength was set at 280 nm. The column temperature was maintained at 40 °C in a thermostatically controlled column compartment. The flow rate was 1.0 mL/min, and the injection volume was 10 µL.

### 2.4. Cell Culture

RAW264.7 murine macrophage and A549 human alveolar epithelial cells were obtained from the Korean Cell Line Bank (Seoul, Korea) and maintained at 37 °C in a CO_2_ incubator (Thermo Fisher Scientific, Waltham, MA, USA). The RAW264.7 and A549 cell lines were maintained in DMEM (Thermo Fisher Scientific) and RPMI (Thermo Fisher Scientific) media supplemented with 10% (*v/v*) fetal bovine serum (Thermo Fisher Scientific) and 1% (*v/v*) penicillin/streptomycin solution (Sigma-Aldrich), respectively [23].

### 2.5. Cell Viability Assay

The cytotoxicity of the CJBs was examined using the WST-8 Cell Viability Kit (Biomax, Seoul, Korea) according to the method of Keum et al. [24], with some modifications. In brief, the RAW264.7 and A549 cells (1 × 10^4^ cells/well) were seeded in 96-well microplates and incubated for 24 h. The cells were then incubated with various concentrations of CJB extract for 24 h (i.e., 0.1, 0.2, 0.5, 1, and 2 mg/mL). Subsequently, the live cells were stained for 2 h with WST-8 dye, and the optical density was read on a microplate reader (Bio-Rad 680; Bio-Rad, Hercules, CA, USA) at 450 nm for the absorbance and at 650 nm for the subtracted background absorbance. The viability was expressed as a percentage of the control.

### 2.6. Intracellular Anti-Oxidative Assay

#### 2.6.1. Nitric Oxide (NO) Analysis

The levels of NO were determined by the method of Park et al. [25], with some modifications. The RAW264.7 cells (2 × 10^5^ cells/well) were plated on a 24-well plate for 24 h, and then 100 ng/mL of lipopolysaccharide (LPS) and CJB solution (i.e., 0.5, 1, and 2 mg/mL) was co-treated for 6 h. The concentration of NO was measured using the Griess reagent. Equal volumes of the supernatant and Griess reagent were mixed (i.e., 50 μL each) and incubated for 15 min at room temperature. The nitrite concentration (μM) was determined by measuring the absorbance at 540 nm using a microplate reader (Bio-Rad) and was compared to a standard curve obtained using sodium nitrite.

#### 2.6.2. Intracellular Levels of ROS

The intracellular levels of ROS were determined by the method of Li et al. [23]. The A549 cells (1 × 10^4^ cells/well) were seeded in a 96-well plate for 24 h, and then 200 μg/mL of PM10 and CJB solution (0.5, 1, and 2 mg/mL) was co-treated for 60 min. A fluorescent probe (2′,7′-Dichlorofluorescin diacetate, 10 μM, Sigma-Aldrich) was then loaded onto the cells and was incubated for 30 min. After the cells were washed twice with ice-cold PBS, they were immediately placed in a spectrofluorometer (Perkin Elmer) to determine the fluorescence intensity of the ROS generation at excitation and emission wavelengths of 535 nm and 485 nm, respectively.

### 2.7. Cell Treatment for Protein Expression

The A549 and RAW264.7 cells (1 × 10^6^ cells/well) were plated in a 60 mm dish for 24 h, and then 200 μg/mL of PM10 or 100 ng/mL of LPS and CJB samples (0.5, 1, and 2 mg/mL) was co-treated for 6 h. After that, the cells were prepared for protein extraction to determine the undergoing mechanism [23].

### 2.8. Animals and Treatment

Twenty-five male seven-week-old BALB/c mice (18–20 g) were purchased from Raon Bio Inc. (Yong-in, Korea) and housed under pathogen-free conditions (24 ± 1 °C, 50 ± 5% humidity and 12 h day/night cycle). During the experiment, the mice had free access to food and water. After a week of acclimation, 25 mice were randomly assigned to five groups: normal control (CON (*n* = 5); saline), PM10 control (PM10 (*n* = 5); PM10 (Sigma-Aldrich) at 10 mg/kg + saline), positive control (NAR (*n* = 5); naringin (Sigma-Aldrich; 95% purity) at 100 mg/kg + PM10); CJB-L (CJB at 100 mg/kg + PM10); and CJB-H (CJB at 200 mg/kg + PM10). Saline, naringin, and CJB were orally administered once daily for one week. On the last day of the NAR and CJB intervention (day 7), 10 mg/kg PM10 was intranasally administered to the PM10, NAR, CJB-L, and CJB-H mice. Saline was nasally administered to the CON mice. After six hours of exposure to PM10, the mice were euthanized by an intraperitoneal injection of tribromoethanol (200 mg/kg; Sigma-Aldrich), and their serum, bronchoalveolar lavage fluid (BALF), and lung tissues were immediately collected and stored at −80 °C [20]. All animal care and experimental procedures were approved by the ethics committee of Korea University (approval number: KUIACUC-2020-0044).

### 2.9. Histological Analysis

The left lobe of the lung tissue was isolated from the mice and immediately fixed in 4% paraformaldehyde before being serially processed and embedded in paraffin. Subsequently, 3 μm tissue sections were prepared using a microtome, and the deparaffinized tissue sections were stained with hematoxylin and eosin (H&E; Sigma-Aldrich). The stained images were visualized under an inverted microscope (Olympus BH 2, Tokyo, Japan; magnification, ×100). The lung injury score was measured on a subjective scale from 1 to 3 as follows: 1, no inflammation was observed; 2, there was occasional cuffing with the inflammatory cells; and 3, most bronchi or vessels were surrounded by inflammatory cells [26].

### 2.10. ELISA Assay

The levels of TNF-α, IL-1β, and IL-6 in BALF were measured using specific ELISA kits (R&D Systems, Minneapolis, MN, USA) according to the manufacturer’s instructions.

### 2.11. Immunoblot Analysis

After exposure to various treatments, the lung tissue and cell extracts were prepared and rinsed in a modified radioimmunoprecipitation assay lysis buffer on ice using the method of Park et al. [25], with slight modifications. After 20 min of centrifugation at 12,000× *g* at 4 °C, the supernatants were collected, and the concentrations were measured using a bicinchoninic acid protein assay kit (Sigma-Aldrich). Subsequently, equal amounts of each protein sample (20 μg) were separated by sodium dodecyl salt-polyacrylamide gel electrophoresis and transferred onto polyvinylidene difluoride membranes (Millipore, Boston, MA, USA) using a Bio-Rad semi-transfer system. After blocking (5% skim milk at room temperature for one hour), the membranes were incubated overnight with primary antibodies (β-actin, pAKT, AKT, pERK, ERK, pJNK, JNK, pNFκB, NFκB, iNOS, COX-2, TNF-α, IL-1β, and IL-6) (Cell Signaling, Danvers, MA, USA; Santa Cruz Biotechnology, Dallas, TX, USA; Abcam, Cambridge, UK)) at 1/1000 or 1/2000 dilutions with agitation at 4 °C. The membranes were then washed and exposed to horseradish peroxidase-conjugated secondary antibodies for one hour at room temperature. Finally, the proteins were visualized using enhanced chemiluminescence reagents (Bio-Rad) and examined with an Image Quant LAS-4000 chemiluminometer (GE Healthcare, Chicago, IL, USA).

### 2.12. Statistical Analysis

The experimental data were processed using IBM SPSS software (version 26.0; IBM Corp, Armonk, NY, USA). The data are expressed as the mean ± standard error from three independent experiments, unless otherwise specified. The differences between the groups were analyzed with one-way analysis of variance followed by Dunnett’s multi-range test, where ^#,^* *p* < 0.05, ^##,^** *p* < 0.01, and ^###,^*** *p* < 0.001 were considered to indicate a significant difference.

## 3. Results

### 3.1. Antioxidant Capacity and Phytochemicals in CJBs

ABTS and DPPH assays are widely used for antioxidant activity determination and are based on a decrease in spectrophotometer absorption at 734 nm and 517 nm due to the reduction of the radicals scavenged by the antioxidant compounds. As shown in Table 1, the CJBs showed free radical scavenging activities with 23.47 ± 0.34 (ABTS) and 28.98 ± 0.42 (DPPH) mg VCE/g of the samples, respectively. The TPC and TFC of the CJBs were analyzed using the Folin–Ciocalteu reagent and the aluminum chloride colorimetric technique, respectively, as shown in Table 1. The TPC values of the extracts ranged from 36.68 ± 0.30 mg GAE/g of the sample. Like the TPC results, the CJBs had a TFC of 21.11 ± 0.49 mg of RE/g of the sample (Table 1). Furthermore, the most abundant bioactive compounds in citrus fruits were analyzed (i.e., naringin and hesperidin) [7] in our study (Figure 1). The quantification data of the naringin and hesperidin content in the CJBs ranged from 4.00 ± 0.09 mg and 9.00 ± 0.03 mg/g of the sample, respectively (Table 1).

### 3.2. Effects of CJB on PM10-Induced Pulmonary Damage

To examine the protective effects of CJBs against PM10-induced lung injuries, changes in lung histopathology were investigated by H&E staining. As shown in Figure 2a, the PM10 group exhibited inflammatory cell infiltration, with cells deposited on the alveolar wall. The lung injury score was calculated following the treatment with naringin and different doses of CJBs (29% reduction at 100 mg/kg and 35% reduction at 200 mg/kg) or NAR (36% reduction at 100 mg/kg), and compared with that of the PM10-treated group, revealing that the naringin and CJBs mitigated inflammatory cell infiltration and protected the lungs from PM10-induced lung damage.

### 3.3. Effects of CJBs on PM10-Induced Pulmonary Inflammation

To understand the preventative mechanisms of CJBs at the molecular level, representative pro-inflammatory cytokines, TNF-α, IL-1β, and IL-6, were analyzed first. In agreement with our histological observations, the secretion of inflammatory cytokines in BALF was dramatically increased by PM10 treatment (710% for TNF-α, 163% for IL-1β, and 570% for IL-6 compared to that the CON group; Figure 3), while the NAR and CJBs (dose-dependent) decreased TNF-α (≈75% for both naringin and CJBs), IL-1β (naringin (38%) and CJBs (31–53%), respectively), and IL-6 (naringin (60%) and CJBs (26–41%), respectively) compared to the PM10 group.

### 3.4. Effects of CJBs on PM10-Induced Signaling Pathways

Particulate pollutants trigger and exacerbate inflammation. In this study, we investigated the regulatory effects of CJBs on PM10-induced pulmonary damage using western blotting. We examined the key proteins involved in major inflammatory signaling pathways: AKT, ERK, JNK, NF-κB, iNOS, COX-2, TNF-α, IL-1β, and IL-6 (Figure 4). The phosphorylation of AKT (336%), ERK (166%), JNK (154%), and NF-κB (203%) was significantly increased in the PM10-treated mice compared to that the CON mice, whereas the CJBs dramatically lowered the protein expression (AKT (44–68%), EKR (20–30%), JNK (27–34%), and NF-κB (13–19%), respectively, compared to the PM10 group) in a dose-dependent manner, similar to the naringin treatment. Additionally, we found that pretreatment with naringin (iNOS (31%), COX2 (13%), TNF-α (26%), IL-1β (16%), and IL-6 (6%)) and CJBs (iNOS (41–46%), COX2 (≈18%), TNF-α (15–38%), IL-1β (≈42%), and IL-6 (40–57%)) in the context of PM10 exposure downregulated the secretion of inflammatory cytokines compared to the PM10 group.

To validate our in vivo study, the same signaling pathway tested in the A549 cells was assessed (Figure 5). Overall, our in vitro study results were in agreement with the in vivo findings; the phosphorylation of AKT (173%), ERK (165%), JNK (264%), and NF-κB (179%) was significantly increased in the PM10 group compared to that in the CON group. The phosphorylated AKT (26–45%), ERK (18–43%), JNK (22–35%), and NF-κB (18–41%) were dramatically decreased in the CJB-treated cells in a dose-dependent manner, regardless of the exposure levels, compared to those in the PM10 group. In addition, the iNOS (28–35%), COX-2 (28–46%), TNF-α (17–33%), IL-1β (23–33%), and IL-6 (30–40%) levels were restored by PM10-induced stimulation in the CJB-treated cells, which is in line with the results concerning lung tissues.

### 3.5. Effects of CJBs on Nitric Oxide Production and Inflammation in RAW264.7 Macrophages

Alveolar macrophages engulf inhaled fine particulates, which can induce pulmonary oxidative stress. To investigate the anti-inflammatory effect of CJBs, the cytotoxicity of CJBs, and LPS-induced NO production was assessed in RAW264.7 macrophages. No significant cytotoxicity was observed (Figure 6a). Additionally, the CJB treatment reduced the NO production in a dose-dependent manner (21–45%) compared to the LPS group (Figure 6b). Furthermore, to better understand the role of CJBs in preventing PM10-induced inflammation, the same oxidation pathways in RAW264.7 macrophages in response to LPS were examined. Like the aforementioned mice and A549 cell studies, the CJBs dose-dependently decreased the phosphorylation of AKT (10–48%), ERK (27–33%), JNK (2–37%), and NF-κB (10–45%) against the LPS treatment in the RAW264.7 macrophages (Figure 7). However, the levels of the other proteins (i.e., iNOS (26–44%), COX-2 (21–36%), TNF-α (18–45%), IL-1β (≈37%), and IL-6 (31–40%)) were decreased in the CJB-treated cells compared to those in the LPS group.

### 3.6. Effects of CJBs on PM10-Induced ROS Generation in A549 Cells

The regulatory effects of the CJBs on PM10-induced intracellular ROS production were measured using a cell-permeable fluorescent probe. Firstly, no significant cytotoxicity of the CJBs was observed (Figure 8a). PM10 exposure dramatically elevated (by approximately 14 times) the intracellular levels of ROS compared to those in the CON group, while the CJB extract significantly reduced (32–45%) ROS generation in a dose-dependent manner compared to the PM10 group, especially at concentrations of 1 and 2 mg/mL (Figure 8b).

## 4. Discussion

Citrus fruit waste, a cost-effective source of bioactive compounds, has pharmacological importance; however, it has not attained commercial importance and is largely disposed of as municipal waste, creating a big challenge for processing industries as well as for pollution-monitoring agencies. Here, we assessed the ability of CJBs, or citrus fruit waste, to inhibit PM10-derived pulmonary damage using in vitro and in vivo approaches. Collectively, the CJBs were effective in reducing PM10-induced lung damage by suppressing pulmonary inflammation, potentially due to their anti-inflammatory and antioxidant properties.

The established health benefits of phytochemicals, which mainly act as antioxidants (e.g., polyphenols and flavonoids) due to their biological activity, necessitate their quantification in food [27,28,29]. The hydrogen-donating ability of the CJB extract was determined using the ABTS and DPPH radical scavenging methods. In addition, the total soluble phenol and flavonoid contents of the CJB extract were determined by the Folin–Ciocalteu assay and by using a method based on the formation of a flavonoid-aluminum complex, respectively. As evidenced in the data reported in Table 1, our results for the ABTS and DPPH assays showed that the CJB extract had a strong radical scavenging capacity equivalent to 23.47 ± 0.83 mg and 28.98 ± 1.03 mg of vitamin C, respectively. Additionally, the CJB extract contained 36.68 ± 0.30 mg of gallic acid, which was equivalent to the total polyphenol content, and 21.11 ± 0.49 mg of rutin, which was equivalent to the total flavonoid content.

The health benefits of citrus fruits have mainly been attributed to the presence of bioactive compounds such as flavonoids (e.g., naringin and hesperidin) [30]. For this reason, the naringin and hesperidin contents were analyzed in this study (Table 1, Figure 1). The naringin and hesperidin contents were identified in the CJB extracts with values of 4.00 ± 0.09 and 9.00 ± 0.03 mg of g sample, respectively. Generally, citrus peels have more abundant flavonoids and a higher content than pulps and juices [31], which is consistent with our results (data not shown). Here, we chose naringin, which has been used mostly in citrus-related research [32,33], as a positive control to investigate its efficacy against PM10-derived pulmonary damage.

PMs are composed of carbon cores and several soluble and insoluble components including acids, organic chemicals, metals, endotoxins, pollen, and fungal debris on their surface [34]. Therefore, the underlying mechanisms by which PM exerts its biological effects are complex. PM exposure may affect many cell types at various levels of immune regulation [35]. Inflammation is considered the central mechanism in the development of various PM-induced pulmonary diseases [36]. Several studies have suggested that endotoxins in PM10 are positively linked to TNF-α, IL-1β, and IL-6 production [37,38]. Likewise, in our experimental conditions, PM10 resulted in histological changes (Figure 2a,b) in mouse lung tissues and increased the inflammatory response in BALF (Figure 3), which is a well-known marker for lung inflammation. On the other hand, the CJB extract treatment significantly prevented PM10-induced histological damage and inflammatory responses (i.e., TNF-α, IL-1β, and IL-6) in BALF, which was enriched with non-cellular and cellular contents (e.g., cytokines and epithelial cells) from the lung alveoli. Thus, elevated levels of inflammatory cytokines in BALF are surrogate markers of lung inflammation.

Oxidative stress is a consequence of excess ROS generation, whereas antioxidant defenses are suppressed. Several studies have supported the notion that oxidative stress is an important mechanism in PM10-induced inflammatory responses, cytotoxicity, and carcinogenesis [39,40]. ROS are generated by various endotoxins (e.g., PM10 and LPS) to activate downstream signaling pathways in order to regulate the inflammatory response. Several studies have demonstrated that PM can trigger several signaling pathways to regulate PM10-induced inflammatory responses, including the AKT, ERK, JNK, and NF-κB signaling pathways [14,41]. The PI3K/AKT signaling pathway is known to play an important role in cell inflammation, apoptosis, angiogenesis, and metastasis. The ERK and JNK signaling pathways are critical intracellular signal-transduction systems and, in general, are activated by a broad array of extracellular and intracellular stimuli. In addition, the NF-κB signaling pathway is a ubiquitous modulator activated by a multitude of stimulants that regulate inflammation, cell cycles, and immune responses. In our study, PM10 stimuli induced ROS production (Figure 8b) and activated the AKT, ERK, JNK, and NF-κB signaling pathways, leading to pulmonary inflammation. We showed that PM10 exposure augmented the phosphorylation of AKT, ERK, JNK, and NF-κB, but the naringin and CJB treatments significantly suppressed the phosphorylation activity in mice (Figure 4). Next, the phosphorylation of AKT, ERK, JNK, and NFκB was assessed in PM10-treated A549 lung cells, which provided evidence that the CJB extract suppressed inflammatory responses via pathways involving AKT, ERK, JNK, and NF-κB (Figure 5). Our corroborating data also showed that downstream inflammatory markers corresponded to AKT, ERK, JNK, and NF-κB activation. iNOS, COX-2, TNF-α, IL-1β, and IL-6 were effectively downregulated by the CJB extract treatment in both the mouse lung tissue and A549 lung cell lines, indicating that the CJB extract is a definite anti-inflammatory agent.

Several studies have reported that PM10 contains endotoxins that act as LPS stimuli to induce inflammation [42,43]. In addition, the activation of macrophages by endotoxins (i.e., LPS) is a major inflammatory induction mechanism [44]. Notably, NO plays a key role in inflammatory responses, and cytokine-activated macrophages trigger NO release [43]. In this study, the CJB extract treatment decreased NO production in the RAW264.7 mouse macrophages in LPS induction (Figure 6b). Moreover, LPS-induced ROS activated AKT, ERK, JNK, and NF-κB phosphorylation, which was highly suppressed by the CJB extract treatment. In addition, the CJB extract treatment suppressed the expression of pro-inflammatory cytokines in the LPS-treated RAW264.7 macrophages (Figure 7), which might have been due to the excessive production of ROS by LPS induction. These findings are consistent with the results obtained in mice and lung cells.

ROS scavenging in cell line models, such as cellular antioxidant assays, was performed because test tube-level assays were not sufficient to prove the biological effects. Fluorescent probe staining was used to determine the intracellular ROS levels in this study (Figure 8). We successfully induced the overproduction of ROS in the A549 cells using PM10, and the CJB extract treatment significantly suppressed ROS levels in a dose-dependent manner for PM10.

## 5. Conclusions

This study demonstrated the prophylactic effects of CJB extract on PM10-induced lung damage and proposed key inflammatory signaling pathways regulated by CJB extract treatment in various models (e.g., in vivo and in vitro). Specifically, we confirmed that CJB treatment effectively prevents PM-induced inflammatory responses by reducing the phosphorylation of AKT, ERK, JNK, and NF-κB and the secretion of cascaded pro-inflammatory cytokines. Furthermore, CJB extract reduces the production of ROS and RNS in a dose-dependent manner. However, there are limitations; this was an acute study, and a high dose of PM10 (10 mg/kg) was chosen to induce the phenotype. Thus, a long-term chronic study might be needed to further confirm the dietary effects as functional ingredients.

## Figures and Tables

**Figure 1 nutrients-14-02270-f001:**
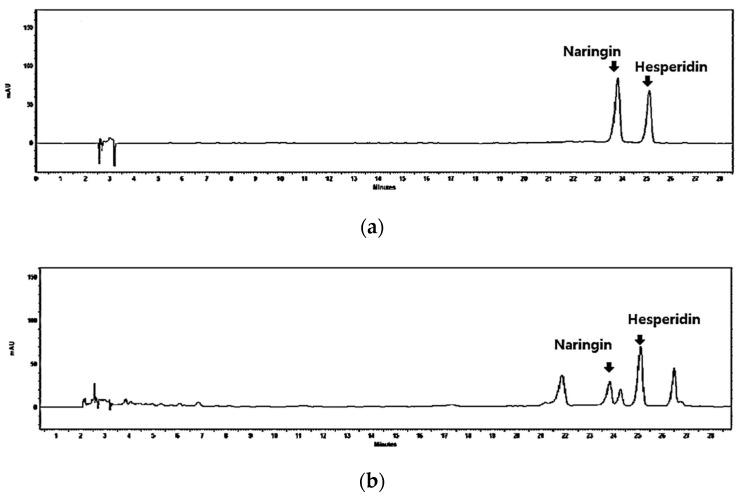
Quantification of naringin and hesperidin from CJBs. Representative high-performance liquid chromatography chromatogram (at 280 nm) of the (**a**) naringin and hesperidin reference standard and (**b**) CJBs. The arrowed peak has been identified as naringin and hesperidin.

**Figure 2 nutrients-14-02270-f002:**
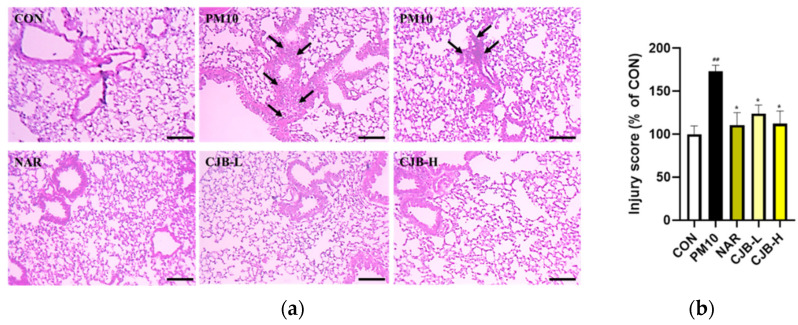
Histological analysis of representative hematoxylin and eosin-stained lung sections and lung pathology assessment of PM10-induced lung damage. (**a**) hematoxylin and eosin-stained sections and (**b**) injury score quantification. Scale bar = 100 μm (magnification, ×100). All values are expressed as the mean ± SEM. A *p*-value under 0.05 was considered statistically significant. ^##^ indicates statistical significance compared to the CON group, and * indicates statistical significance compared to the PM10 group. CON, normal control; PM10, negative control; NAR, naringin at 100 mg/kg; CJB-L, CJB at 100 mg/kg; CJB-H, CJB at 200 mg/kg.

**Figure 3 nutrients-14-02270-f003:**
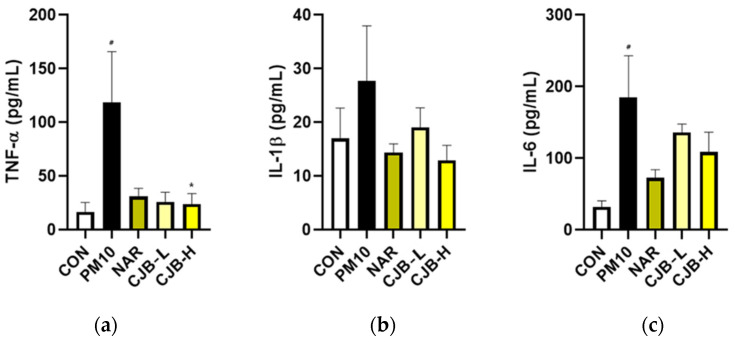
The expression of pro-inflammatory cytokines in the BALF from PM10-treated mice. (**a**) TNF-α, (**b**) IL-1β, and (**c**) IL-6 protein expression. All values are expressed as the mean ± SEM. A *p*-value under 0.05 was considered statistically significant. ^#^ indicates statistical significance compared to the CON group, and * indicates statistical significance compared to the PM10 group. CON, normal control; PM10, negative control; NAR, naringin at 100 mg/kg; CJB-L, CJB at 100 mg/kg; CJB-H, CJB at 200 mg/kg.

**Figure 4 nutrients-14-02270-f004:**
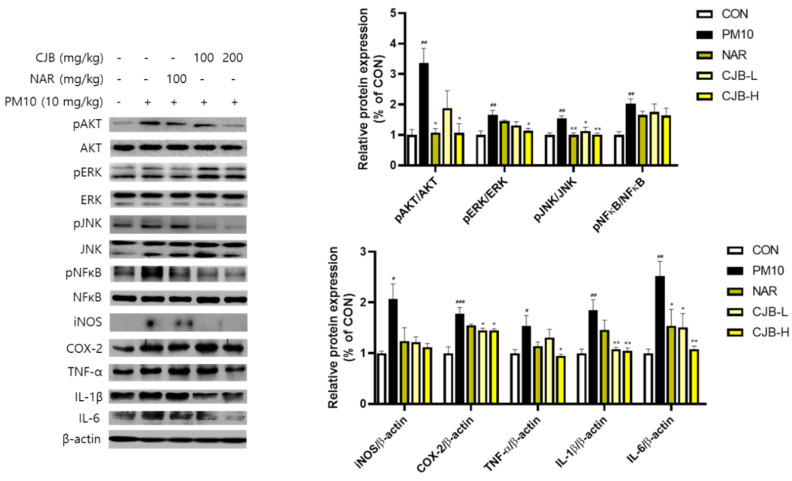
The expression of inflammatory proteins in the lung tissue from PM10-treated mice. All values are expressed as the mean ± SEM. A *p*-value under 0.05 was considered statistically significant. ^#^, ^##^, and ^###^ indicate statistical significance compared to the CON group, and * and ** indicate statistical significance compared to the PM10 group. CON, normal control; PM10, negative control; NAR, naringin at 100 mg/kg; CJB-L, CJB at 100 mg/kg; CJB-H, CJB at 200 mg/kg.

**Figure 5 nutrients-14-02270-f005:**
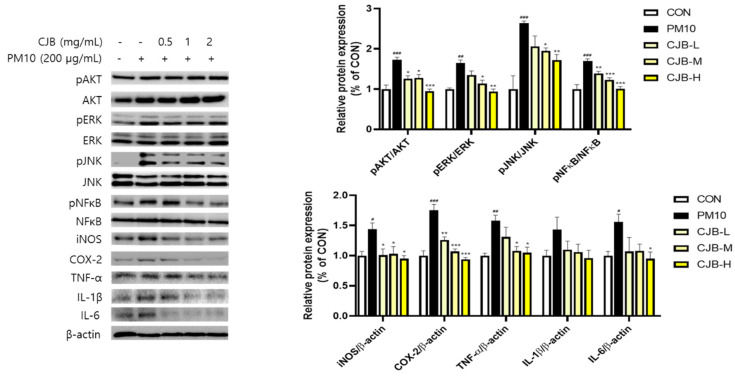
The expression of inflammatory proteins in PM10-treated A549 cell lines. All values are expressed as the mean ± SEM. A *p*-value under 0.05 was considered statistically significant. ^#^, ^##^, and ^###^ indicate statistical significance compared to the CON group, and *, **, and *** indicate statistical significance compared to the PM10 group. CON, normal control; PM10, negative control; CJB-L, CJB at 0.5 mg/mL; CJB-M, CJB at 1 mg/mL; CJB-H, CJB at 2 mg/mL.

**Figure 6 nutrients-14-02270-f006:**
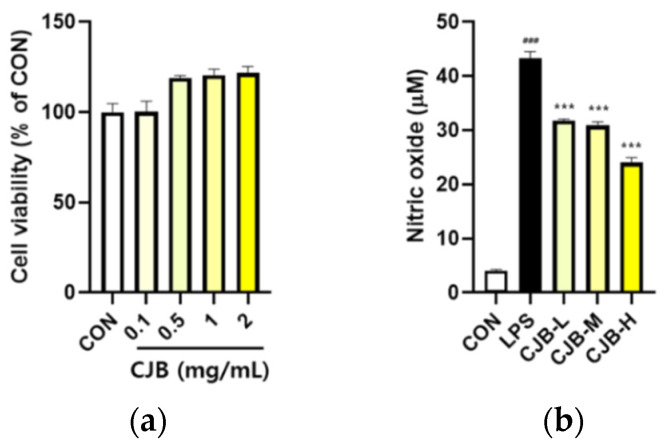
Cell cytotoxicity and production levels of nitric oxide (NO) in LPS-treated mice. (**a**) Cytotoxicity and (**b**) NO concentration. All values are expressed as the mean ± SEM. A *p*-value under 0.05 was considered statistically significant. ^###^ indicates statistical significance compared to the CON group, and *** indicates statistical significance compared to the LPS group. CON, normal control; LPS, negative control; CJB-L, CJB at 0.5 mg/mL; CJB-M, CJB at 1 mg/mL; CJB-H, CJB at 2 mg/mL.

**Figure 7 nutrients-14-02270-f007:**
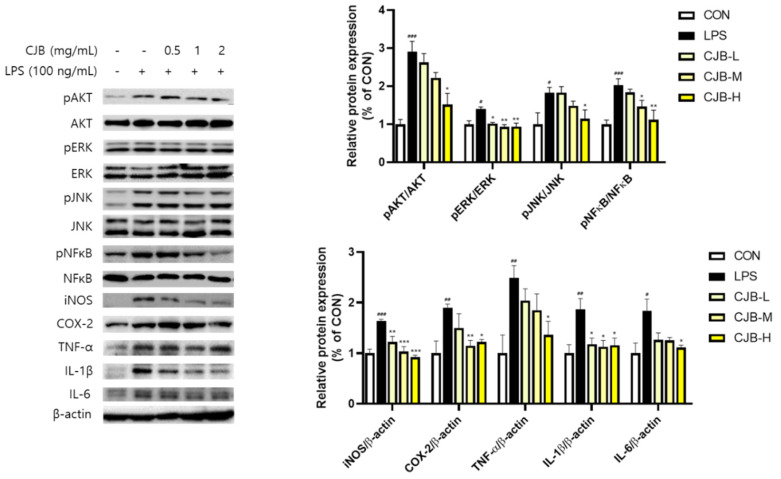
The expression of inflammatory proteins in LPS-treated RAW264.7 cell lines. All values are expressed as the mean ± SEM. A *p*-value under 0.05 was considered statistically significant. ^#^, ^##^, and ^###^ indicate statistical significance compared to the CON group, and *, **, and *** indicate statistical significance compared to the LPS group. CON, normal control; LPS, negative control; CJB-L, CJB at 0.5 mg/mL; CJB-M, CJB at 1 mg/mL; CJB-H, CJB at 2 mg/mL.

**Figure 8 nutrients-14-02270-f008:**
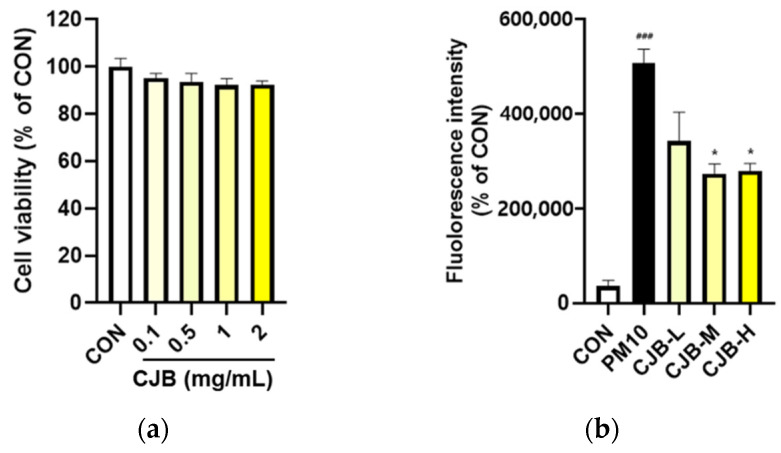
Cell cytotoxicity and production levels of intracellular ROS in PM10-treated A549 cell lines. (**a**) Cytotoxicity and (**b**) intracellular antioxidant activity. All values are expressed as the mean ± SEM. A *p*-value under 0.05 was considered statistically significant. ^###^ indicates statistical significance compared to the CON group, and * indicates statistical significance compared to the PM10 group. CON, normal control; PM10, negative control; CJB-L, CJB at 0.5 mg/mL; CJB-M, CJB at 1 mg/mL; CJB-H, CJB at 2 mg/mL.

**Table 1 nutrients-14-02270-t001:** Radical scavenging capacity and phytochemicals in CJBs.

ABTS Assay	DPPH Assay	Total Polyphenol Content	Total Flavonoid Content	Naringin	Hesperidin
mg VCE/g sample		mg GAE/g sample	mg RE/g sample	mg/g sample/retention time (min)	
23.47 ± 0.34	28.98 ± 0.42	36.68 ± 0.30	21.11 ± 0.49	4.00 ± 0.09 (23.8)	9.00 ± 0.03 (25.1)

The data are expressed as the mean ± SEM. VCE, vitamin C equivalent; GAE, gallic acid equivalent; RE, rutin equivalent.

## Data Availability

Not applicable.

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
