# Peer review of "Ameliorative Effect of Citrus junos Tanaka Waste (By-Product) Water Extract on Particulate Matter 10-Induced Lung Damage"

_nutrients, 2022, doi:10.3390/nu14112270_

Round 1
Reviewer 1 Report
Dear Editor,
The manuscript nutrients-1692778, “Ameliorative effect of Citrus junos Tanaka waste (by-product) water extract on particulate matter 10-induced lung damage”, represents an interesting study on industrial waste of Citrus junos that may provide a valuable health-relevant recycling. The study design and data presenting are fine. It could be considerably improved by a better reviewing bibliography of related studies on Citrus junos in the introduction. Every experiment must be explained in the materials and methods. Discussion is mostly describing the results again. Some other points that need to be reconsidered are:
- How the cells/animals were treated should be briefly mentioned in the abstract.
- What is heat-distilled water? Is it the same as hot distilled water?
- Line 89 “The rate of absorbance at 517 nm was measured using a spectrophotometer (Perkin Elmer) and the results are presented for the VCE mg/g sample.” Needs rewording.
- How did the authors select the PM10 and CJB extract doses, 200 µg/ml and 0.5, 1, and 2 mg/ml?
- Line 145: what was the fluorescence probe loaded onto the cells?
- How the A549 and RAW264.7cells were treated?
- Part 3.4. please check the downregulated values of the secretion of inflammatory cytokines.
- Part 3.4.: How were A549 cells treated with PM10?
- The x axis in figure3 6a must be defined.
- Line 325: What was “a cell permeable fluorescence probe”?
- The manuscript needs a careful proofread.
Best regards,
Author Response
Point 1: How the cells/animals were treated should be briefly mentioned in the abstract.
Response 1: We appreciate the reviewer’s comment. As suggested, we added following brief information regarding cells and animal treatment conditions (page 1, lines 18~20).
☞ In this study, we investigated the ameliorative effects of CJB extracts (100, 200 mg/kg/day, 7 days) on PM10-induced (10 mg/kg, intranasal, 6 hours) lung damage in BALB/c mice. Cell type-specific signaling pathways are examined using the A549 (PM10, 200 μg/mL, 6 hours) and RAW264.7 (LPS, 100 ng/mL, 6 hours) cell lines.
Point 2: What is heat-distilled water? Is it the same as hot distilled water?
Response 2: We appreciate the reviewer's comments. As aforementioned, we revised the extraction condition on page 2, line 71.
☞ The material was ground using a blender and extracted (70°C) with 10 volumes of distilled water for two hours.
Point 3: Line 89 “The rate of absorbance at 517 nm was measured using a spectrophotometer (Perkin Elmer) and the results are presented for the VCE mg/g sample.” Needs rewording.
Response 3: The authors thank you for the comment. We revised the wording on line 92.
☞ The rate of absorbance at 517 nm was monitored using a spectrophotometer (Perkin Elmer) and the results are presented as mg VCE/g sample.
Point 4: How did the authors select the PM10 and CJB extract doses, 200 µg/ml and 0.5, 1, and 2 mg/ml?
Response 4: We appreciate the reviewer's thoughtful comments. As for the concentration of CJB extract, three concentrations (0.5, 1, and 2 mg/mL) that effectively removed intracellular ROS without showing cytotoxicity were selected (Figure 8). On the other hand, for the PM10, although the reference data was not shown, we determined the PM10 condition in which NF-κB phosphorylation and intracellular ROS were dramatically increased without noticeable cell toxicity. The reference data was shown below.
Point 5: Line 145: what was the fluorescence probe loaded onto the cells?
Response 5: The probe is 2′,7′-Dichlorofluorescin diacetate, normally used to demonstrate intracellular ROS levels (line 153).
Point 6: How the A549 and RAW264.7cells were treated?
Response 6: We appreciate the reviewer's thoughtful comments. As the reviewer commented, we added the detailed information about A549 and RAW264.7 cells treatment conditions in the method section 2.7 (line 158).
☞ 2.7. Cell treatment for protein expression
The A549 and RAW264.7 cells (1×106 cells/well) were plated in a 60 mm dish for 24 hours, and then 200 μg/mL of PM10 or 100 ng/mL of LPS, and CJB samples (0.5, 1, and 2 mg/mL) were co-treated for 6 hours. After then, the cells were prepared for protein extraction to determine the undergo mechanism.
Point 7: Part 3.4. please check the downregulated values of the secretion of inflammatory cytokines.
Response 7: We have checked the values and the mentioned markers are properly described.
Point 8: Part 3.4.: How were A549 cells treated with PM10?
Response 8: We appreciate the reviewer's thoughtful comment. We have added the detailed information about the A549 and RAW264.7 cells treatments in method section 2.7 (line 158).
Point 9: The x axis in figure3 6a must be defined.
Response 9: As commented, we have revised the figure 6a and figure 8a.
Point 10: Line 325: What was “a cell permeable fluorescence probe”?.
Response 10: The probe we utilized was the 2′,7′-Dichlorofluorescin diacetate, normally used to demonstrate intracellular ROS levels.
Point 11: The manuscript needs a careful proofread.
Response 11: We appreciate the reviewer’s thoughtful comments. We have already received English proofreading services, and we would like to submit the editing certificate below.

Reviewer 2 Report
Comments to the Author
The paper pertains to Ameliorative effect of Citrus junos Tanaka waste (by-product) water extract on particulate matter 10-induced lung damage. It is an interesting study; however, I suggest a major revision of the paper according to the following comments.
1-The manuscript should be carefully revised for typos and grammatical errors
2- The introduction section need rewrite with recent reference
3- The initial weight of the mice was not defined. Please, add this information.
4- Describe on which base you select the dose of treatment and duration
5- The author should include the scale bars on all histopathological photographs
6- Add reference to the method of extraction
7- All measurements of these parameters need a reference (Total Flavonoid Content, Analysis of Naringin and Hesperidin Content, Cell Culture, Cell Viability Assay, Nitric oxide (NO) analysis)
Author Response
Point 1: The manuscript should be carefully revised for typos and grammatical errors
Response 1: We appreciate the reviewer’s thoughtful comment. We have received the professional English proofreading service and thus provided the certificate from the company.
Point 2: The introduction section need rewrite with recent reference.
Response 2: Thanks for the reviewer's thoughtful comment. In response to the comment, we have updated references in the introduction.
Point 3: The initial weight of the mice was not defined. Please, add this information.
Response 3: As the reviewer commented, we added the initial weight of the mice on line 164.
Point 4: Describe on which base you select the dose of treatment and duration.
Response 4: We appreciate the reviewer's meaningful comment. The design of this animal study, the dose, and duration for treatment of PM10 and CJB, were justified in our previously published paper. The author added the reference paper in the method section of 2.8 (line 177).
Point 5: The author should include the scale bars on all histopathological photographs
Response 5: We appreciate the reviewer’s comment. As the reviewer commented, we have added the scale bars, and the detailed information was included in Figure 2 legend (line 253).
☞ Figure 2. Histological analysis of representative hematoxylin & eosin-stained lung sections and lung pathology assessment of PM10-induced lung damage. (A) hematoxylin & eosin-stained sections and (B) injury score quantification. Scale bar=100 μm (magnification, ×100). All values are expressed as mean ± SEM. A p-value under 0.05 was considered statistically significant. ## indicates statistical significance compared to the CON group and * indicates statistical significance compared to the PM10 group. CON, normal control; PM10, negative control; NAR, naringin at 100 mg/kg; CJB-L, CJB at 100 mg/kg; CJB-H, CJB at 200 mg/kg.
Point 6: Add reference to the method of extraction
Response 6: As commented, we have added the reference for the sample extraction (line 70).
Point 7: All measurements of these parameters need a reference (Total Flavonoid Content, Analysis of Naringin and Hesperidin Content, Cell Culture, Cell Viability Assay, Nitric oxide (NO) analysis)
Response 7: We appreciate the reviewer’s comments. As suggested, we have added the references to respective parameters and included them in the material and methods section.
☞ Total Flavonoid Content – line 95; Analysis of Naringin and Hesperidin Content – line 111; Cell Culture – line 128; Cell Viability Assay – line 131; Nitric oxide (NO) analysis – line 141

Round 2
Reviewer 2 Report
Accept